# Access to Mental Health and Substance Use Resources for 2SLGBTQ+ Youth during the COVID-19 Pandemic

**DOI:** 10.3390/ijerph182111315

**Published:** 2021-10-28

**Authors:** Michael Chaiton, Iman Musani, Mari Pullman, Carmen H. Logie, Alex Abramovich, Daniel Grace, Robert Schwartz, Bruce Baskerville

**Affiliations:** 1Dalla Lana School of Public Health, University of Toronto, Toronto, ON M5T 3M7, Canada; Iman.musani@mail.utoronto.ca (I.M.); Alex.Abramovich@camh.ca (A.A.); daniel.grace@utoronto.ca (D.G.); Robert.Schwartz@utoronto.ca (R.S.); 2Institute for Mental Health Policy Research, Centre for Addiction and Mental Health, Toronto, ON M6J 1H4, Canada; mari.pullman@camh.ca; 3Factor-Inwentash Faculty of Social Work, University of Toronto, Toronto, ON M5S 1A1, Canada; carmen.logie@utoronto.ca; 4Department of Psychiatry, University of Toronto, Toronto, ON M5S 1A1, Canada; 5Canadian Institutes of Health Research, Ottawa, ON K1A 0W9, Canada; Bruce.Baskerville@cihr-irsc.gc.ca; 6School of Pharmacy, Faculty of Science, University of Waterloo, Kitchener, ON N2G 1C5, Canada

**Keywords:** 2SLGBTQ+, COVID-19, youth, pandemic, mental health, substance use, gender minority, sexual minority

## Abstract

Previous research has established that gender and sexual minority (2SLGBTQ+) youth experience worse mental health and substance use outcomes than their heterosexual and cisgender counterparts. Research suggests that mental health and substance use concerns have been exacerbated by the COVID-19 pandemic. The current study used self-reported online survey responses from 1404 Canadian 2SLGBTQ+ youth which included, but were not limited to, questions regarding previous mental health experiences, diagnoses, and substance use. Additional questions assessed whether participants had expressed a need for mental health and/or substance use resources since the beginning of the COVID-19 pandemic (March 2020) and whether they had experienced barriers when accessing this care. Bivariate and multinomial logistic regression analyses were conducted to determine associations between variables and expressing a need for resources as well as experiencing barriers to accessing these resources. Bivariate analyses revealed multiple sociodemographic, mental health, and substance use variables significantly associated with both expressing a need for and experiencing barriers to care. Multinomial regression analysis revealed gender identity, sexual orientation, ethnicity, and level of educational attainment to be significantly correlated with both cases. This study supports growing research on the mental health-related harms that have been experienced during the COVID-19 pandemic and could be used to inform tailored intervention plans for the 2SLGBTQ+ youth population.

## 1. Introduction

The COVID-19 pandemic has had enormous mental health impacts on populations worldwide, with some groups being more vulnerable to poor mental health outcomes than others [1]. Previous research has established that youth who identify as Two-Spirit, lesbian, gay, bisexual, transgender, queer, and questioning (2SLGBTQ+), hereafter also referred to as youth belonging to gender and sexual minority groups, experience mental health concerns at a disproportionately higher rate than their cisgender and/or heterosexual counterparts [2]. While any individual seeking mental health resources may experience general barriers to access, gender and sexual minority individuals often experience additional barriers, which are impediments to care due to stigma against these individuals [3,4]. Alarmingly, recent research has found gender and sexual minority youth may disproportionately experience mental health-related harms as a result of the COVID-19 pandemic, including heightened anxiety, depression, substance misuse, and suicidality [1,5,6]. In a retrospective case series examining psychiatric hospitalizations stemming from the COVID-19 pandemic in Los Angeles, California, LGBTQ2S+ youth who were hospitalized for mental health concerns stated that lack of family acceptance during shelter-in-place orders significantly impacted their mental well-being [6]. These sentiments were echoed in another study that was conducted through Q Chat Space, an online professionally facilitated support group center for gender and sexual minority youth operating out of the United States. [7] These youth cited lack of support and inability to utilize established coping mechanisms previously provided by the health care system as reasons for declining mental health during the initial wave of the COVID-19 pandemic [7].

In addition to experiencing greater mental health challenges, there is increasing concern about the ability for 2SLGBTQ+ youth to access quality mental health care during the ongoing COVID-19 pandemic. A report published by the Trevor Project highlighted the immense barriers faced by 2SLGBTQ+ youth in accessing appropriate mental health services in the United States [8]. Existing barriers to access for gender and sexual minority youth include medical mistrust and discrimination from frontline providers. As such, 2SLGBTQ+ youth consistently report lower rates of health care engagement [9]. These barriers have largely been exacerbated in the face of the current pandemic. One explanation for this poorer access is an increased burden on all aspects of the health care system due to COVID-19. As a result of this heightened demand, health care priorities have shifted, leaving children and youth to be deprioritized in many cases [10]. Along with this shift in priorities, stay-at-home orders and physical distancing guidelines have impacted day-to-day functioning in schools and facilities providing social services [5,11]. Research suggests that schools are among the most prominent mental health support providers for 2SLGBTQ+ youth, due in part to stigma many of these youth face in their home environments as well as additional barriers to access for external resources [11]. In-school resources typically include counselling, gender and sexuality alliances, and social supports (e.g., peers, coaches, and teachers) [11]. In the absence of these resources and the inability of schools to provide consistent, in-person support, youth are left without the mental health support they may have previously relied on [5,11].

While there has been a substantial increase in the provision of virtual telehealth services for mental health, 2SLGBTQ+ youth continue to face challenges accessing these services. In a recent study investigating mental health services in the context of COVID-19, 2SLGBTQ+ participants were less than half as likely to utilize telehealth mental health services compared to their heterosexual counterparts [9]. There are several proposed reasons for this underutilization of resources and decreased access to care. In many instances, youth who live in unsupportive home environments have reported feeling unsafe using phone-based services, due to fear of being overheard by family members [7,9]. These youth have also described the feeling of being isolated from their “chosen families” and community supports during the pandemic, an important aspect of support they previously received. They noted that this aspect of peer and community support has not yet been translated to the telehealth services they were being offered. Through testimonials from 2SLGBTQ+ youth, the importance of text-based resources and integrated peer support during the COVID-19 pandemic were heavily emphasized [7].

There have also been limitations noted regarding the nature of care that can be provided through virtual platforms. Notably, for transgender and non-binary youth who are seeking gender-affirming care and treatments (i.e., hormone therapy, transition-related surgeries), there have been considerable disruptions in the access to services [11]. Transgender and non-binary youth have reported increased gender dysphoria as a direct result of being unable to access the necessary gender-affirming treatments [11]. Gender dysphoria has several negative mental health implications, including heightened depression and suicidality [12,13].

Finally, 2SLGBTQ+ youth who belong to other marginalized groups (i.e., those from lower socioeconomic strata, youth experiencing homelessness) are additionally disadvantaged in their access to care because they are likely to face challenges accessing the reliable, private technology and internet connection necessary to effectively utilize telehealth services [14]. These youth also face additional challenges in relation to COVID-19, including high susceptibility to the virus due to dense living and working conditions, as well as lack of access to other medical supports, including treatments for substance use disorders (e.g., methadone treatments, safe consumption sites) and sexually transmitted infections [14]. These challenges may have impacts on the mental health of vulnerable 2SLGBTQ+ youth and create additional barriers in their ability to access mental health supports during this time.

The current analysis sought to further understand the barriers faced by 2SLGBTQ+ youth to accessing mental health and addiction services during the COVID-19 pandemic. We also aimed to reveal common characteristics of youth who reported experiencing barriers to accessing services with the goal of informing appropriate and accessible services.

## 2. Materials and Methods

### 2.1. Data Source

Data for this analysis were taken from the LGBTQI2S Tobacco Project Screening Questionnaire which surveyed youth between November 2020 and March 2021. Ethics approval for the current study was granted by the University of Toronto. Participants were recruited using paid advertisements through Facebook, Instagram, and Grindr. Additionally, individuals within the networks of youth working group members, advisory group members, and young adult advisors for the project were also recruited through snowball sampling. Investigators used a quota sampling protocol with the goal of the final sample including 900 tobacco smokers and 600 non-smokers. Half of the participants were recruited from three pilot communities: Toronto, Thunder Bay, and Montreal, while the other half were recruited from across Ontario and Quebec; the survey was provided in both English and French. Informed consent was collected online prior to participation in this study. A total of 1500 youth completed the survey and were compensated $10 CAD each for their participation. Descriptive statistics were calculated for all variables included in the analysis.

### 2.2. Variables

The outcome measure for this analysis was the self-reported experience of barriers to access or delay in accessing mental health or addiction services since March 2020, based on questions developed by Burgess from the Behavioural Risk Factor Surveillance System [15]. The outcome variable was binary and comprised of (1) youth who expressed a need for mental health or addiction services since March 2020 and experienced barriers or delays in accessing these services and (2) youth who expressed a need for mental health or addiction services since March 2020 and did not experience barriers or delays in accessing these services. Individuals were classed based on questions (1) “Since March 2020, was there a time when you wanted to talk with or seek help from a health professional about stress, depression, problems with emotions or substance use?” and (2) “Did you delay or not get the care you thought you needed?” Seeking help from a health professional would include any and all forms of aid including telehealth services. Participants who responded “yes” to the first question and “yes” to the second were classed as having expressed a need for resources and experienced barriers. Those who responded “yes” to the first question and “no” to the second were classed as having expressed a need for resources and not experienced barriers. Individuals who answered “no” to the first question were those who did not express a need for resources to begin with. Additionally, both questions allowed individuals to state “I don’t know” or “Prefer not to answer”.

Sociodemographic variables included in the analysis were language the survey was conducted in, level of educational attainment, time of residence in Canada (years), individual income, household income, type of municipality (rural/urban), age (years), gender identity, current sexual orientation, and ethnicity.

Mental health variables included were a lifetime diagnosis of any mental illness (yes/no), current experience of depression (assessed by the short form of the Centre for Epidemiological Studies Depression (CES-D) scale embedded into the survey) [16], suicidal ideation in the past year, and diagnoses of any of 15 individual mental health conditions in a “yes or no” format (anorexia or bulimia, anxiety disorder, attention deficit disorder (ADD), attention deficit hyperactivity disorder (ADHD), bipolar disorder, depression, dysthymia, mania, obsessive compulsive disorder (OCD), panic disorder, phobia, psychosis, post-traumatic stress disorder (PTSD), schizophrenia, and any mental disorder not captured by previous categories).

Substance use variables were also explored as frequency of use in the past year. Substance variables of interest included alcohol (daily use, 4 times a week, 2–3 times a week, once a week, 2–3 times a month, once a month, less than once a month, or not at all), cannabis (same frequency measures as alcohol), cigarette smoking (daily or almost daily, less than daily but at least weekly, less than weekly but at least monthly, less than monthly, or not at all), e-cigarette use (same frequency measures as cigarette smoking), and illicit drug use (any use in the past year). The illicit drugs included in this analysis were poppers/amyl, ketamine, ecstasy/MDMA, crystal meth, crack, cocaine, heroin, other prescription opioids (e.g., Percocet, Dialudid, OxyContin), fentanyl, GHB, tranquilizers or benzodiazepines, psychedelics, and any other illicit drug not captured by the aforementioned categories.

In addition to these sociodemographic, mental health, and substance-related variables, we included participants’ self-reported ratings of their physical and mental health. These measures were collected separately in the survey and were measured using a 5-point Likert scale from “Poor” to “Excellent”.

### 2.3. Bivariate Analysis

Bivariate analyses were conducted between the outcome variable and each demographic, mental health, and substance use variable included in the sample. For categorical variables of interest, chi square analyses were conducted as cross-tabulations with the outcome variable of barriers to access. For continuous variables in the sample, analyses of variance (ANOVA) were conducted. A significance threshold of α = 0.05 was used for all bivariate tests. All analyses were conducted using SAS (SAS Institute Inc., Cary, NC, USA) [17] and Stata (StataCorp, College Station, TX, USA) [18].

### 2.4. Regression Analyses

A multinomial logistic regression was conducted on the demographic variables in predicting barriers to accessing mental health/substance use care. The multinomial logistic regression was used to find distinct demographic correlates for individuals who either (a), did not express a need for resources, (b), did express a need for resources and did not face barriers, and (c), did express a need for resources and did face barriers. Again, a significance threshold of α = 0.05 was used for the logistic regression analyses and 95% confidence intervals are outlined as well.

## 3. Results

### 3.1. Descriptive Statistics

Descriptive statistics were calculated for sociodemographic, mental health, and substance use variables included in the analysis. Of the original 1500 individuals surveyed, 96 respondents either stated “I don’t know” or “Prefer not to answer” or did not provide a response to one or both of the two questions that were used to assess the outcome measures noted above. These individuals were excluded from subsequent analyses and data from 1404 participants were analyzed. A majority of survey respondents reported experiencing a need for mental health/addiction services in the past year and facing barriers when accessing them (*n* = 815; 58.1%). 21.4% (*n* = 300) of respondents reported experiencing a need for mental health/addiction services and being able to access these resources without barriers. Finally, 20.6% (*n* = 289) stated they did not experience a need for any mental health/addiction services within the past year.

The average age of participants included in this analysis was 21.9 years (*SD* = 3.78), ranging from ages 15 to 29 years. The majority of respondents opted to conduct the survey in English (*n* = 1142, 81.3%), were White (*n* = 996, 74.3%) and had resided in Canada for their entire lives (*n* = 1149, 85.9%). In terms of educational attainment, 34.6% of respondents (*n* = 459) had a university degree or higher. Participants reported a range of gender identities, 47.1% of respondents identified their current gender as women (*n*= 643), 23.7% identified as men (*n* = 323), 28.4% (*n* = 388) of these youth identified with gender expansive identities (e.g., gender non-conforming, non-binary, genderqueer, genderfluid) and 0.7% (*n* = 10) identified as Two-Spirit. Participants also identified with a range of sexual orientations, 29.9% identified as bisexual (*n* = 420), 16.7% identified as queer (*n* = 234), 16.6% as lesbian (*n* = 233), 16.0% as gay (*n*= 225), 12.6% as pansexual (*n*= 177), 4.6% as asexual (*n* = 65), 1.1% as heteroflexible (*n*= 16), 0.5% as Two-Spirit (*n* = 7), 0.4% as straight/heterosexual (*n* = 6), and 1.5% of individuals stated that they were unsure or questioning (*n* = 21). Descriptive statistics and bivariate analyses for all demographic variables analyzed can be found in Table 1. In all following tables, 1–5, categories in which the total sample number does not equal 1404 participants is due to individuals selecting “I don’t know” or “Prefer not to answer” in response or leaving the variable measure blank.

### 3.2. Bivariate Analysis

#### 3.2.1. Demographic Variables

Of the demographic variables discussed above, age (*p* = 0.002), language in which the survey was conducted (*p* = 0.001), gender identity (*p* < 0.001), sexual orientation (*p* < 0.001), ethnicity (*p* < 0.001), level of educational attainment (*p* < 0.001), and both individual (*p* < 0.001) and household income (*p* < 0.001) were found to be associated with expressing a need for mental health and addiction services. On average, younger individuals, English-speaking individuals, and lower-income individuals (both household and individual) were more likely to have sought services. Variables of interest within the scope of the current study and the direction of these associations with respect to expressing a need for and experiencing barriers to accessing mental health and addiction services are further investigated below under the regression analysis.

#### 3.2.2. Mental Health Variables

Most (69.2%) participants (*n* = 967) reported that their mental health was either poor or fair on a five-rating scale ranging from poor to excellent. Using the same scale measuring for general health, 67.6% of participants (*n* = 936) reported that their health was either fair or good. 62.3% of participants (*n* = 861) had reported experiencing suicidal ideation within the past year and 59.9% reported that they had received a prior mental health diagnosis (*n* = 797). Self-perceived mental health (*p* < 0.001), self-perceived general health (*p* < 0.001), suicidal ideation within the past year (*p* < 0.001), and any mental health diagnosis (*p* < 0.001) were found to be significantly associated with expressing a need to access mental health and addiction services all generally in an inverse direction.

Regarding the above-mentioned variables and specific mental health disorders, both descriptive statistics and bivariate analyses can be found in Table 2. 8.8% of respondents reported an anorexia/bulimia diagnosis within their lifetime (*n* = 123), 43.4% reported anxiety disorder (*n* = 609), 6.7% reported ADD (*n* = 94), 11.4% reported ADHD (*n* = 160), 5.1% reported bipolar disorder (*n* = 75), 37.8% reported depression (*n* = 531), 2.6% reported dysthymia (*n* = 37), 1.4% reported mania (*n* = 20), 7.4% reported OCD (*n* = 104), 9.5% reported panic disorder (*n* = 134), 2.2% reported a phobia disorder (*n* = 31), 1.9% reported psychosis (*n* = 26), 12.0% reported PTSD (*n* = 168), 0.1% reported schizophrenia (*n* = 1), and 10.8% of respondents reported another mental health diagnosis that was not one of the options provided (*n* = 152). Among these mental health diagnoses, anorexia or bulimia (*p* = 0.003), anxiety disorder diagnoses (*p* < 0.001), ADD (*p* = 0.009), ADHD (*p* < 0.001), bipolar (*p* = 0.024), depression (diagnosed (*p* < 0.001) and as measured by the embedded CES-D scale (*p* < 0.001)), dysthymia (*p* = 0.022), OCD (*p* < 0.001), panic disorder (*p* < 0.001), phobia *(p* = 0.038), PTSD (*p* < 0.001), and other mental health diagnoses (*p* < 0.001) were associated with expressing a need for services. In all significant associations, a mental health diagnosis increased the likelihood of the individual expressing a need for services. 

#### 3.2.3. Substance Use Variables

Among variables used to measure substance use, 13.3% of individuals reported daily tobacco use (*n* = 186), 9.8% reported vaping daily (*n* = 138), 2.4% reported daily alcohol use (*n* = 33), and 12.2% reported daily cannabis use (*n* = 171). Current tobacco smoking frequency (*p* = 0.004), alcohol use frequency (*p* = 0.017), and cannabis use frequency (*p* = 0.001) were all associated with the experience of barriers to mental health and addiction services. In addition, any use of illicit substances (*p* = 0.006) within the past year was associated with barriers to access. Within the sample, 31.2% of participants (*n*= 438) reported using some form of illicit substance within the past year. 3.6% of individuals reported using poppers or amyl (*n* = 50), 2.7% reported using ketamine (*n* = 38), 8.8% reported using ecstasy or MDMA (*n* = 123), 1.1% reported using crystal meth (*n* = 15), 1.0% reported using crack (*n* = 14), 10.2% reported using cocaine (*n* = 143), 1.0% reported using heroin (*n* = 14), 6.0% reported using some form of other prescription drug (n = 84), 1.2% reported using fentanyl (*n* = 17), 1.3% reported using GHBs (*n* = 18), 6.9% reported using tranquilizers or benzodiazepines (*n* = 97), 18.4% reported using psychedelics (*n* = 259), and 2.8% reported using some other illicit substance not listed above over the past year (*n* = 40). Specifically, experiencing barriers to accessing mental health and addiction services was associated with the use of tranquilizers/benzodiazepines (*p* = 0.031) and psychedelic drugs (*p* < 0.001). All data on substance use variables and bivariate analysis can be found in Table 3.

### 3.3. Regression Analysis

A multinomial logistic regression was conducted to determine significant distinctions in demographic variables associated with any of the three situations regarding accessing care: (a), those who did not express a need for resources, (b), those who did express a need for resources and did not face barriers when accessing them, and (c), those who did express a need for resources and did face barriers when accessing them. The analysis identified differences in gender identity, sexual orientation, ethnicity, and level of education to be significant correlates for expressing a need for care as well as experiencing barriers to accessing this care. Data from the multinomial logistic regression can be found in Table 4 and Table 5.

#### 3.3.1. Expressing a Need for Care

In comparing correlates that distinguished individuals who did express a need for care from those who did not, gender, sexual orientation, ethnicity, and education were found to be significant. Certain gender identity and sexual orientation categories were combined to provide greater statistical power. The researchers believed that any variable with less than 20 participants would be too insignificant to hold statistical power in the context of the broader sample. For gender, 10 individuals identified as Two-Spirit and were therefore combined with the next smallest label, gender non-conforming (*n* = 48) to create a new label with a larger sample size (*n* = 58). For sexual orientation, heteroflexible (*n* = 16), straight/heterosexual (*n* = 6), and Two-Spirit *(n* = 7) were combined into one label (*n* = 29) for the same reason.

Those who identified as gender non-binary (OR 3.08, 95% CI 1.59, 6.00), gender non-conforming/Two-Spirit (OR 2.89, 95% CI 1.05, 7.94), or as women (OR 1.61, 95% CI 1.01, 2.48) were significantly more likely to express a need for care than those who identified as men. Those who identified as pansexual (OR 1.99, 95% CI 1.01, 3.95) were more likely to express a need for care than those who identified their sexual orientation as bisexual. Conversely, those who identified as gay (OR 0.43, 95% CI 0.25, 0.73) were significantly less likely to express a need for care than those who identified as bisexual. Furthermore, those who reported to be mixed-race (OR 2.88, 95% CI 1.17, 7.09) were more likely than White participants to have expressed a need for care. However, South Asian participants (OR 0.44, 95% CI 0.20, 0.97) were less likely than White participants to have expressed this need. Additionally, University educated individuals (OR 0.43, 95% CI 0.22, 0.85) were found to be less likely to have expressed a need for care than those who did not complete high school.

#### 3.3.2. Expressing a Need for Care and Experiencing Barriers

Comparing participants who had expressed a need for resources and did face barriers when accessing them to those who did not express a need for resources revealed that gender identity, sexual orientation, and ethnicity were significantly associated. Individuals who identified as gender non-binary (OR 3.51, 95% CI 1.66, 7.46) and gender non-conforming/Two-Spirit (OR 3.48, 95% CI 1.12, 10.81) reported experiencing barriers to access at a significantly higher rate than those who identified as men. Those who identified as pansexual (OR 2.22, 95% CI 1.05, 4.71) were significantly more likely to report experiencing barriers to resources than those who identified as bisexual. However, those who identified as gay (OR 0.44, 95% CI 0.23, 0.84) and lesbian (OR 0.52, 95% CI 0.30, 0.92) were significantly less likely to report experiencing these barriers compared to those who identified as bisexual. Furthermore, mixed-race individuals (OR 3.87, 95% CI 1.48, 10.08) also reported experiencing barriers to accessing these resources in comparison to White participants.

#### 3.3.3. Experiencing Barriers to Access

A post hoc analysis conducted to determine significant distinctions in individuals who did face barriers as opposed to individuals who did not face barriers in accessing care revealed that education was the only significant association. Specifically, those who were college/ Collège d’enseignement général et professionnel (CEGEP) educated (or something of equivalent experience) (OR = 0.51, 95% CI 0.26, 0.99) or university educated (OR = 0.42, 95% CI 0.21, 0.48) were significantly less likely than those who were less educated to face barriers when trying to access care.

## 4. Discussion

### 4.1. Implications for Research

The goal of this analysis was to better understand common characteristics of 2SLGBTQ+ youth who have experienced barriers to accessing mental health services in Canada during the COVID-19 pandemic. Our exploratory analysis sought to reveal areas for future study and potential focus areas for mental health services directed at gender and sexual minority youth. To date, little research has investigated the barriers to accessing mental health services among this heterogeneous group in a Canadian context [7].

Our results support an emerging body of research suggesting that gender and sexual minority youth are experiencing immense mental health challenges during the current COVID-19 pandemic [8,11,14]. Approximately 80% of youth who responded to the current survey stated that they felt the need to seek mental health and/or addiction supports since March 2020. Further, the majority of respondents (62.3%) expressed experiencing some form of suicidal ideation in the past year. A study conducted in 2019, prior to the COVID-19 pandemic, reported that 42% of the gender and sexual minority youth surveyed had accounts of suicidal ideation [19]. Although the 2019 study asked respondents about suicidal ideation within the last six months as opposed to the last year as was asked in the current study, it may still suggest that the COVID-19 pandemic has exacerbated thoughts of suicide. In addition to a high proportion of respondents reporting mental health challenges, our results reinforced research which suggested that gender and sexual minority youth faced substantial barriers to accessing mental health and addiction services during this time, with a majority of respondents stating that they experienced barriers to accessing mental health or addiction services since March 2020 [12,20]. Some of these mental health and substance use impacts may be experienced more broadly, but the survey responses do suggest that the COVID-19 pandemic has served to exacerbate some of these effects.

Our analysis revealed some of the sociodemographic, mental health, and substance-related factors that vary among youth who express a need for mental health and/or addiction services and are unable to adequately access such services. Notably, we observed that youth who reported facing barriers to access varied significantly based on socioeconomic indicators including income (both household and individual), and education level. This finding is in line with research suggesting that low socioeconomic status is associated with decreased access to mental health services for youth, both within and beyond the context of the COVID-19 pandemic [21,22]. In addition to socioeconomic indicators, we observed that the age of participants was associated with barriers to access. This association should be investigated further, as age often correlates with other important characteristics such as income and educational attainment, especially between the ages of 15 and 29 [23,24].

Multiple sociodemographic variables were found to be significant correlates both for expressing a need for care and experiencing barriers in finding care. Differences in gender identity, sexual orientation, ethnicity, and education were found to be significantly associated in distinguishing individuals who expressed a need for care from those who did not. Furthermore, distinctions in gender identity, sexual orientation, and ethnicity were significant correlates in distinguishing those who experienced barriers when seeking care from those who did not express a need for care.

#### 4.1.1. Expressing a Need for Care

Those who identified as gender non-binary, gender non-conforming/Two-Spirit and women were more likely to express a need for care than those who identified as men. This observation may be due to societal pressure to not express a need for mental health resources that men experience, which has been well researched [25,26]. Previous research has identified that men are often pressured to conform to outdated gender norms regarding masculinity and may feel that expressing a need or seeking help for mental health opposes this expectation [25]. Individuals who identified as pansexual were more likely to express a need for care than those who identified as bisexual. However, those who identified as gay were less likely to express a need for care than those who identified as bisexual. This pattern may be explained by past research on sexual orientation and the binarity/monosexuality associated with these categories [27]. Individuals who identified with non-monosexual labels, such as bisexual or pansexual, may experience discrimination due to this and may find difficulty assimilating with a group. This may lead to frustration and isolation in youth that affects mental health, leading them to seek resources for help [27].

Regarding other sociodemographic factors correlated with expressing a need for care, mixed-race individuals were more likely than White participants to have expressed this need whereas South Asians were less likely to have. Previous studies on racial identity have mirrored the above research on less categorical sexual identities. Mixed-race individuals may struggle with mental health in youth due to added identity frustration because of their mixed heritage which may serve to exacerbate any existing conditions [28]. Furthermore, in a study focusing on American military personnel, the researchers found that amongst all racial groups studied (Asian, White, Black, Indigenous and Other), Asian Americans were the least likely to utilize available mental health resources [29]. This suggests that cultural differences in how expressing a need for and/or seeking mental health resources is perceived may account for South Asians attempting to access these resources at a decreased rate.

Lastly, university educated individuals were less likely than their counterparts who had not completed high school to have expressed a need for these resources. This may be socioeconomic in nature, those who can afford university typically come from wealthier backgrounds which tend to have lower rates of mental illness [30]. Specifically, with regard to the COVID-19 pandemic, factors related to lower education such as scarce job opportunities, especially those in which individuals can work from home, and less access to reliable information regarding COVID-19, may act as an additional stressor exacerbating underlying mental health conditions [31].

#### 4.1.2. Expressing a Need for Care and Experiencing Barriers

Distinguishing those who did express a need for care and did face barriers from those who did not express a need for these resources, those who identified as gender non-binary and gender non-conforming/Two-Spirit reported experiencing barriers to access at a greater rate than those who identified as men. Furthermore, those who identified as pansexual were most likely to have expressed a need for resources followed by those who identified as bisexual, then as gay, similar patterns can be observed in individuals who experience barriers to accessing care. Those who identified as pansexual were much more likely to report experiencing barriers to resources followed by those who identified as bisexual, then those who identified as gay and lesbian. Those who identify with a non-monosexual or non-unigender label may find resources and services to be limited because they have not been expanded to be inclusive and affirming to all individuals within the population [32]. Although mental health and substance use services may be 2SLGBTQ+ affirming and inclusive, they should cater to all individuals within the population and should receive ongoing and mandatory 2SLGBTQ+ inclusion training and education to ensure this.

Furthermore, mixed-race individuals reported experiencing barriers to accessing these resources in comparison to White participants. Similar to resources that may exist that focus on specific gender and sexual minority individuals, those with less categorical or mixed racial identities may find that resources that cater specifically to them are limited. For mixed-race individuals with a blend of cultural identities and influences, they may find difficulty in accessing resources that are inclusive to their identity. Our findings on barriers experienced by those who are sexual and gender diverse and/or of racial minority backgrounds highlights the importance of finding culturally appropriate mental health services.

Lastly, findings revealed that university, college, and CEGEP educated individuals were less likely to experience barriers than those who had not obtained a high school diploma. Again, this may be socioeconomic in nature, while higher educated individuals are less likely to express a need for resources in the first place, when they do, they may have the financial resources to find the appropriate supports without barriers.

Regarding mental health diagnoses, we found that mental health status over the past year (both as a diagnosed mental disorder and as a self-reported current mental health rating) was associated with experiencing barriers to accessing services. When examining specific mental health diagnoses, we noted that a majority of individual mental disorder diagnoses were associated with experiencing barriers to accessing services (including, but not limited to, anxiety, depression, anorexia or bulimia, and OCD; full results can be seen in Table 2). This finding supports a strong body of research suggesting that different mental illnesses are associated with varying levels of stigma, social acceptability, and treatment options [33]. Moreover, youth belonging to gender and sexual minority groups face unique experiences of mental health that intersect with their gender and sexual identities [1,5,10]. As such, despite an expedited introduction of many telehealth-based services seeking to address mental health concerns for these young people, it is evident that current services may not address the complexity necessary to adequately support gender or sexual minority youth.

In addition to diagnosed mental illness, we observed that any suicidal ideation in the past year was associated with barriers to accessing mental health/addiction services. This finding is of particular concern. Young people who belong to gender and sexual minority groups frequently report suicidal ideation at much higher rates than their cisgender and heterosexual counterparts [34]. A recent study suggested that feelings of community connectedness among 2SLGBTQ+ youth is an important protective factor against suicidal ideation and the progression of poor mental health [34]. The understanding that much of the community connectedness youth were experiencing before the pandemic has been removed due to physical distancing requirements, along with the heightened prevalence of suicidal ideation among these groups in general, points to the need for mental health services to be accessible and sufficient for youth in these groups [5,7,14,34].

Finally, we examined substance use variables and found that frequency of smoking cigarettes, using cannabis, and consuming alcohol were associated with experiencing barriers to accessing mental health/addiction services. Notably, we did not observe any association between frequency of e-cigarette (vaping) use and the experience of barriers to access. The use of legal and widely available substances, such as alcohol, cigarettes, and cannabis, has been found to occur alongside mental health challenges and mental disorders. [35,36,37,38] Given this high degree of co-occurrence, it is possible that the relationship between barriers to accessing services and the use of legal substances are due to their association with the presence of a mental illness.

Regarding the use of illicit substances, we found that overall use of illicit substances was found to be associated with the experience of barriers to access, as well as the use of tranquilizers/benzodiazepines and psychedelic substances in the past year. This finding is of concern because these substances have potential to cause a great degree of harm on their own, as well as when they exist alongside other substance use and mental health challenges [39]. In addition to these harms, recent studies have noted that gender or sexual minority youth have reported using substances more often during the COVID-19 pandemic [11,40]. The nature of this association between substance use and access to mental health/addiction services must be investigated further.

### 4.2. Implications for Practice

Our findings on suicidal ideation and the need for mental health services within this community are significant and serve as a call to action. Similar findings documenting unmet needs within the 2SLGBTQ+ population and subpopulatins within the community have been previously documented [41,42]. The striking figures regarding mental health disruptions experienced by the 2SLGBTQ+ youth population (59.9% of our population reported a previous mental health diagnosis), especially in comparison to the general youth population in Canada, of which approximately 20% have a mental health disorder [43], highlights how mental health services must be tailored to where the greatest needs lie. Previous studies that surveyed members of the 2SLGBTQ+ community on where they believe health care services are lacking suggested that sexual minority affirming care needs to be emphasized; many believed that care is often heterocentric and “one-size-fits-all”, disregarding the experiences of sexual minority patients [3,44]. Service providers should recognize that gender and sexual minority individuals may have had previous experiences with health care providers in which they may have been discriminated against or stigmatized; these individuals may continue to feel distrust when approaching service providers and the providers should work to lessen these hesitations [3]. The current study successfully identified that increased mental health disruptions experienced by 2SLGBTQ+ youth can be attributed, at least in some part, to barriers that are experienced when attempting to access the appropriate services. The researchers encourage future studies to identify where these barriers experienced by the 2SLGBTQ+ youth population specifically lie. 

### 4.3. Limitations

Although the current study provides valuable insight into the need for mental health and substance use resources experienced by gender and sexual minority youth, there are several limitations that should be mentioned. While the survey was distributed throughout a variety of social spheres, the sample was skewed to represent more White, urban dwelling, and University educated individuals. This may have caused some imbalance in the representation of individuals from other backgrounds within the sample. It should also be noted that there were 22 respondents that identified as straight, heterosexual, or heteroflexible. This is not enough people to make a comparison against the 2SLGBTQ+ respondents within the sample. Therefore, the current study cannot conclude that 2SLGBTQ+ individuals experience mental health and/or substance use related harms and challenges at a greater rate in comparison to their cisgender and heterosexual counterparts, although other studies mentioned above do suggest that this seems evident.

As with any study of this nature, the inability to infer causation from correlational research must be noted. The sociodemographic and mental health/substance use variables associated with experiencing a need for mental health/substance use services and/or experiencing barriers to accessing these services cannot be seen as causal factors. It may be the case that some of these variables are associated with an overarching factor that is more strongly associated with these two cases.

Additionally, although studies have determined that many of the services offered to 2SLGBTQ+ individuals have been limited due to the COVID-19 pandemic, and it can possibly be inferred that this would heighten the need for mental health/substance use care, the current study did not have a baseline assessment that was conducted prior to the beginning of the pandemic. Rather, a retrospective approach was used to assess the need for and barriers to accessing 2SLGBTQ+ youth mental health and/or substance use resources since the beginning of COVID-19 pandemic. Therefore, participant recall regarding these variables may potentially be inaccurate and subject to biases.

## 5. Conclusions

Overall, our findings support ongoing and emerging studies from other regions which have suggested that the COVID-19 pandemic has led to disproportionate mental health-related harms for vulnerable groups, including 2SLGBTQ+ youth. Furthermore, the current study revealed that a majority of the respondents not only expressed experiencing a need for mental health and addictions services after March 2020, but also faced barriers when attempting to access them. This analysis provided a valuable basis for future studies in a Canadian context to further investigate commonalities between individuals who are experiencing the mental health impacts of the COVID-19 pandemic. Future studies should clarify the nature of these associations in order to understand and address challenges faced by youth who are especially vulnerable to the immediate and long-term impacts of the current COVID-19 pandemic.

## Figures and Tables

**Table 1 ijerph-18-11315-t001:** Cross-Tabulations of Demographic Variables.

Variable		*N*	Expressed Need for Help, Faced Barriers (%)	Expressed Need for Help, Did Not Face Barriers (%)	Did Not Seek Help (%)	Total (%)	*p*
Age		1404	58.1	21.4	20.6	100%	0.002
(*M* = 21.9,							
*SD* = 3.78)							
Language	English	1142	84.5	78.0	75.8	81.3	0.001
	French	262	15.5	22.0	24.2	18.7	
Gender Identity	Genderfluid	53	4.2	4.8	2.1	3.9	<0.001
	Genderqueer	62	4.2	5.8	4.2	4.5	
	Gender non-binary	225	18.5	19.9	7.4	16.5	
	Gender non-conforming /Two-Spirit	58	4.6	5.1	2.5	4.3	
	Man	323	19.4	19.9	39.6	23.7	
	Woman	643	49.2	44.5	44.2	47.1	
Orientation	Asexual	65	4.0	5.7	5.2	4.6	<0.001
	Bisexual	420	31.9	30.7	23.5	29.9	
	Gay	225	11.4	13.0	32.2	16.0	
	Straight/ Heterosexual/ Heteroflexible/ Two-Spirit	29	2.1	2.7	1.7	2.1	
	Lesbian	233	16.7	13.0	20.1	16.6	
	Pansexual	177	13.7	16.0	5.9	12.6	
	Queer	234	18.8	18.3	9.0	16.7	
	Not sure/ Questioning	21	1.5	0.7	2.4	1.5	
Ethnicity	East Asian	48	3.4	2.5	5.1	3.6	<0.001
	South Asian	51	2.8	4.9	5.5	3.8	
	Southeast Asian	26	2.6	1.1	1.1	1.9	
	Black— African/ Caribbean/ North American	38	2.0	2.1	5.9	2.8	
	First Nations/ Indigenous/ Inuit/Metis	35	2.9	1.8	2.6	2.6	
	Latin American	32	2.2	2.1	3.3	2.4	
	Middle Eastern	27	2.4	1.8	1.1	2.0	
	White—North American	686	49.2	50.7	57.5	51.2	
	White—Other	310	25.3	25.0	15.0	23.1	
	Mixed heritage	87	7.2	8.1	2.9	6.5	
Environment	Urban (100,000+ people)	892	66.9	69.6	62.2	66.5	0.187
	Medium city/town (30,000–99,999 people)	279	19.3	19.9	25.8	20.8	
	Small city/town (1000–29,999 people)	131	10.6	7.3	9.9	9.8	
	Rural (<1000 people)	40	3.2	3.1	2.1	3.0	
Education	Less than high school	161	14.1	9.8	9.1	12.1	<0.001
	High school diploma	227	19.1	13.7	15.2	17.1	
	Some post-secondary	250	20.5	18.9	14.1	18.8	
	Registered apprenticeship/trades certificate or diploma	18	1	0.7	2.9	1.4	
	College, CEGEP, other	212	16.2	18.2	13	16.0	
	University degree	459	29.1	38.6	45.7	34.6	
Individual Income	Less than $15,000	648	60.9	54.1	35.5	54.2	<0.001
	$15,000 to $29,000	260	20.8	25.3	20.8	21.8	
	$30,000 to $59,999	186	14.1	16	19.2	15.6	
	$60,000 to $79,000	75	3.2	2.3	19.2	6.3	
	More than $80,000	26	1	2.3	5.3	2.2	
Household Income	Less than $15,000	131	13.9	14.7	7.6	12.7	<0.001
	$15,000 to $29,000	177	18.8	18.4	11.6	17.2	
	$30,000 to $59,999	221	23.4	21.7	16	21.4	
	$60,000 to $79,000	174	13.2	13.8	29.3	16.9	
	$80,000 to $100,000	118	10	8.3	18.2	11.4	
	More than $100,000	210	20.5	23	17.3	20.4	
Length of time in Canada	2 years or less	39	2.6	3.5	3.2	2.9	0.551
	3 to 5 years	36	2.5	1.7	4.3	2.7	
	6 to 10 years	44	3.2	3.1	3.6	3.3	
	11 to 20 years	70	5.3	3.8	6.4	5.2	
	All my life	1,149	86.4	87.8	82.5	85.9	
**Total**		1404	58.1	21.4	20.6	100%	

**Table 2 ijerph-18-11315-t002:** Cross-Tabulations of Mental Health Variables.

Variable		*N*	Expressed Need for Help, Faced Barriers (%)	Expressed Need for Help, Did Not Face Barriers (%)	Did Not Seek Help (%)	Total (%)	*p*
Self-perceived mental health	Poor	467	40.6	34.4	12.2	33.4	<0.001
	Fair	500	39.5	39.1	21.9	35.8	
	Good	272	16.2	19.1	29.2	19.5	
	Very good	132	3.6	6.7	28.8	9.4	
	Excellent	26	0.1	0.7	8	1.9	
Self-perceived general health	Poor	122	10.7	9.2	3.2	8.8	<0.001
	Fair	422	35.4	29.6	17.5	30.5	
	Good	514	36.4	39.8	36.5	37.1	
	Very good	262	14.9	17.0	32.3	18.9	
	Excellent	64	2.6	4.4	10.5	4.6	
Suicidal ideation (any in the past year)	Yes	861	72.7	67.7	27	62.3	< 0.001
Any mental health diagnosis	Yes	797	65.5	77.2	25.2	59.9	<0.001
Depression (CES-D; *M* = 20.05, *SD* = 5.16)							<0.001
ADD	Yes	94	7.0	9.3	3.1	6.7	0.009
ADHD	Yes	160	13.0	14.7	3.5	11.4	<0.001
Anorexia/bulimia	Yes	123	9.8	10.7	3.8	8.8	0.003
Anxiety disorder	Yes	609	49.2	55.0	14.9	43.4	<0.001
Bipolar disorder	Yes	72	5.6	6.7	2.1	5.1	0.024
Depression	Yes	531	42.8	48.3	12.8	37.8	<0.001
Dysthymia	Yes	37	2.7	4.3	0.7	2.6	0.022
Mania	Yes	20	1.8	1.3	0.3	1.4	0.181
OCD	Yes	104	8.3	10.7	1.4	7.4	<0.001
Panic Disorder	Yes	134	11.7	9.7	3.5	9.5	<0.001
Phobia	Yes	31	2.0	4.0	1.0	2.2	0.038
Psychosis	Yes	26	1.7	3.3	0.7	1.9	0.054
PTSD	Yes	168	13.9	14.7	3.8	12	<0.001
Schizophrenia	Yes	1	0	0.3	0	0.1	0.159
Other mental health diagnosis	Yes	152	11.3	17.3	2.8	10.8	<0.001
**Total**		1404	58.1	21.4	20.6	100%	

**Table 3 ijerph-18-11315-t003:** Cross-Tabulations of Substance Use Variables.

Variable		*N*	Expressed Need for Help, Faced Barriers (%)	Expressed Need for Help, Did Not Face Barriers (%)	Did Not Seek Help (%)	Total (%)	*p*
Alcohol use	None	210	13.0	14.3	21.1	15.0	0.017
	Daily	33	2.1	2.7	2.8	2.4	
	Less than daily	1161	84.9	83.0	76.1	82.7	
Cannabis use	None	512	34.2	36.0	43.3	36.5	0.001
	Daily	171	14.1	13.3	5.5	12.2	
	Less than daily	721	51.7	50.7	51.2	51.4	
Tobacco use	None	927	66.1	68.3	63.8	66.1	0.004
	Daily	186	12.5	9.3	19.5	13.3	
	Less than daily	289	21.3	22.3	16.7	20.6	
Vaping	None	1022	72.9	74.0	71.3	72.8	0.543
	Daily	138	10.4	9.7	8.3	9.8	
	Less than daily	244	16.7	16.3	20.4	17.4	
Past year illicit substance use (overall)	Yes	438	32.6	34.7	23.5	31.2	0.006
Past year cocaine	Yes	143	10.4	10.3	9.3	10.2	0.867
Past year crack	Yes	14	0.7	1.3	1.4	1.0	0.511
Past year crystal meth	Yes	15	1.1	0.3	1.7	1.1	0.254
Past year ecstasy/ MDMA	Yes	123	10.1	7.3	6.6	8.8	0.121
Past year fentanyl	Yes	17	1.0	0.7	2.4	1.2	0.098
Past year GHB	Yes	18	1.1	1.0	2.1	1.3	0.400
Past year heroin	Yes	14	1.0	0.3	1.7	1.0	0.233
Past year ketamine	Yes	38	3.3	2.0	1.7	2.7	0.252
Past year poppers/amyl	Yes	50	3.6	5.0	2.1	3.6	0.160
Past year other prescription drugs	Yes	84	7.1	5.3	3.5	6.0	0.069
Past year psychedelics	Yes	259	20.2	22.0	9.7	18.4	<0.001
Past year tranquilizers/benzos	Yes	97	6.9	9.7	4.2	6.9	0.031
Past year other substances	Yes	40	3.2	3.7	1.0	2.8	0.106
**Total**		1404	58.1	21.4	20.6	100%	

**Table 4 ijerph-18-11315-t004:** Redictors of seeking help for mental health and substance use issues **without** barriers or delays to access among 2SLGBTQ+ youth and youth adults (16–29) Canada, 2021.

Variables		Odds Ratio	*p*	95% Confidence Interval	
Age		1	0.999	0.944	1.06
Gender	Man	1	.	.	.
	Genderfluid	2.189	0.153	0.748	6.402
	Genderqueer	1.085	0.851	0.465	2.533
	Gender non-binary	3.075	0.001 **	1.585	5.966
	Gender non-conforming/ Two-Spirit	2.885	0.04 *	1.048	7.944
	Woman	1.611	0.045 *	1.01	2.481
Orientation	Asexual	1			
	Bisexual	0.572	0.136	0.272	1.193
	Gay	0.428	0.002 **	0.25	0.73
	Straight/Heterosexual/ Heteroflexible/Two-Spirit	0.803	0.716	0.247	2.616
	Lesbian	0.634	0.053	0.403	1.007
	Pansexual	1.991	0.048 *	1.005	3.946
	Queer	1.609	0.1.5	0.906	2.857
	Not sure/Questioning	0.547	0.288	0.18	1.664
Ethnicity	White	1	.	.	.
	East Asian	1.081	0.842	0.502	2.338
	South Asian	0.439	0.042 *	0.198	0.971
	Southeast Asian	2.251	0.296	0.492	10.3
	Black—African/Caribbean/ North American	0.454	0.051	0.206	1.002
	First Nations/ Indigenous/Inuit/Metis	0.855	0.754	0.32	2.285
	Latin American	0.517	0.171	0.201	1.328
	Middle Eastern	2.034	0.289	0.548	7.553
	Mixed heritage	2.878	0.022 *	1.169	7.09
Environment Population		0.868	0.155	0.714	1.055
Education	Less than high school	1			
	High school diploma	0.766	0.392	0.416	1.411
	Some post-secondary	0.906	0.764	0.477	1.722
	Registered apprenticeship/ trades certificate or diploma	0.391	0.165	0.104	1.469
	College, CEGEP, other	0.844	0.631	0.423	1.684
	University degree	0.43	0.015 *	0.217	0.849
Length of time in Canada		1.147	0.101	0.974	1.352

** *p* < 0.01, * *p* < 0.05.

**Table 5 ijerph-18-11315-t005:** Predictors of seeking help for mental health and substance use issues **with** barriers or delays to access among 2SLGBTQ+ youth and youth adults (16–29) Canada, 2021.

Variables		Odds Ratio	*p*	95% Confidence Interval	
Age		0.999	0.959	0.932	1.069
Gender	Man	1			
	Genderfluid	2.256	0.186	0.675	7.544
	Genderqueer	1.322	0.584	0.488	3.582
	Gender non-binary	3.514	0.001 **	1.656	7.455
	Gender non-conforming/ Two-Spirit	3.479	0.031 *	1.119	10.81
	Woman	1.436	0.229	0.796	2.59
Orientation	Asexual	1			
	Bisexual	0.867	0.738	0.375	2.001
	Gay	0.438	0.014 *	0.227	0.844
	Straight/Heterosexual/ Heteroflexible/Two-Spirit	1.288	0.704	0.349	4.76
	Lesbian	0.521	0.024 *	0.296	0.918
	Pansexual	2.222	0.037 *	1.049	4.706
	Queer	1.361	0.354	0.71	2.612
	Not sure/Questioning	0.153	0.091	0.017	1.352
Ethnicity	White	1			
	East Asian	0.648	0.42	0.226	1.861
	South Asian	0.824	0.672	0.335	2.024
	Southeast Asian	0.827	0.854	0.11	6.215
	Black—African/Caribbean/ North American	0.452	0.125	0.164	1.247
	First Nations/ Indigenous/Inuit/Metis	0.516	0.318	0.141	1.892
	Latin American	0.557	0.335	0.169	1.832
	Middle Eastern	2.085	0.336	0.467	9.317
	Mixed heritage	3.864	0.006 **	1.482	10.075
Environment Population		0.809	0.081	0.638	1.027
Education	Less than high school	1			
	High school diploma	0.937	0.867	0.435	2.018
	Some post-secondary	1.377	0.429	0.626	3.015
	Registered apprenticeship/ trades certificate or diploma	0.436	0.377	0.069	2.752
	College, CEGEP, other	1.648	0.244	0.711	3.82
	University degree	0.991	0.983	0.429	2.289
Length of Time in Canada		1.154	0.155	0.947	1.406

** *p* < 0.01, * *p* < 0.05.

## Data Availability

Anonymized data can be requested from the corresponding author from investigators at academic institutions with research ethics board approval.

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
