# Peer review of "Access to Mental Health and Substance Use Resources for 2SLGBTQ+ Youth during the COVID-19 Pandemic"

_ijerph, 2021, doi:10.3390/ijerph182111315_

Round 1
Reviewer 1 Report
This is an interesting and very well-written paper looking at exploring the barriers faced by 2SLGBTQ+ youth to accessing mental health and addiction services during the COVID-19 pandemic. Although I think this paper should be published, I would suggest authors some minor revisions, as follows:
- Please, add some descriptive comments to the Results paragraph “Demographic variables”. Authors just wrote that some variables were associated with barriers to accessing mental health and addiction services. Specifically, authors should add some explanation of the direction of the association (e.g., what happens to the barriers to accessing mental health at the increase/decrease of age?)
- The same is for the paragraph “Mental health variables”.
- The discussion is thorough and examines all findings. However, I came out a bit confused as many results are discussed together. I suggest authors to create subparagraphs.
- I would like to read some implications for social policies and public health.
Reviewer 2 Report
A very well designed and well written study. The introduction provides an excellent overview of current literature as well as defining terms used throughout this research. It highlights the need for improved research strategies and the current lack of information around access to care in pandemic situations. The methods are very well described with appropriate data collection and data analysis tools. The results are presented well, although Table 1 and Table 4 are quite large and can be difficult to interpret over 2+ pages. I would suggest reducing the font size to help a little. Also, check spacing in line 278. The results align with the aims of the study and the discussion demonstrates high level thinking and good application of the findings from this study. The utilisation of relevant and recent literature helps bring the work together. The concluding statements summarise the research well and it is great to see a limitations section.
Author Response
The results are presented well, although Table 1 and Table 4 are quite large and can be difficult to interpret over 2+ pages. I would suggest reducing the font size to help a little.
- Thank you for your suggestion, Table 1 has been simplified slightly (with smaller font and some variables reduced) and the labels above have been copied over to the 2nd page so readers do not have to flip back and forth between pages. It has been reduced from 3+ pages to 2 which we believe simplifies it enough but still provides all of the necessary data to readers.
- Table 4 has been split into 2 which should make it very easy to digest.
Also, check spacing in line 278.
- This has been corrected, thank you.
Reviewer 3 Report
Thank you for the opportunity to review this manuscript! The topic is highly relevant and with multiple implications. The manuscript is well written and the language is appropriate. The findings have important implications. I only have a few suggestions: 1. In the abstract, the sentence "Multinomial regression analysis revealed gender identity, sexual orientation, ethnicity, and level of educational attainment to be significantly correlated with both cases" seems reductionist considering all the gender identity and sexual orientation options. The authors need to highlight conducting a study with all these options. 2. On page 16 all statistical symbols need to be italicized. There are other parts where they are not italicized appropriately. 3. In the Introduction, the authors addressed the relationship between telehealth services and sexual orientation and gender identity. However, in the questions the participants are asked, it is not clear if the barriers they are asked about were for this type of service. I think this needs clarification in the manuscript. 4. The authors collapsed some of the sexual orientation and gender identity options. Have they tested to see if there are any differences between these categories? Could they say more about their process and their decision making of which ones to select?
Reviewer 4 Report
Thank you for the opportunity to review ‘Access to Mental Health and Substance Use Resources for 2SLGBTQ+ Youth During the COVID-19 Pandemic’. The data on which the study is based are drawn from a tobacco project screening questionnaire, and collected from 1500 participants between November 2020 and March 2021. The study aims to assess the impact of Covid-related lockdowns on access to mental health services for gender and sexually diverse young people, ages 15-29.
The study is well-conceived, well-designed, and carefully analysed; there is a limitations section that appropriately identifies recruitment and other methodological concerns. Overall the paper is well-written. I commend the authors for these things. Nevertheless, there are a few questions that emerged for me as I read the paper. The authors note that the original sample was 1500, but that only 1404 (line 192) respondents were included in this study. What happened to the other 96? What were the inclusion (or exclusion) criteria for this analysis? The table totals (which are not included, but should be the last row for each category) do not add up to 1404—for instance in the Gender section the totals add only to 1364. What accounts for the missing data? Was this no-response or missing, ‘other’, or something else?
In lines 280-21 the paper notes that certain categories were compressed to provide greater statistical power; this is certainly understandable (we’ve all done it), but what were the criteria?
There is no mention of language in the paper, yet (and I must note that I am not Canadian) participants appear to have been recruited largely from Francophone Canada. What were the language(s) of the survey, and if it was available in both English and French, were there differences by primary language? If the survey was not available in both languages, should this be noted in the limitations section?
In the regression analysis, section 3.2.4 (lines 266ff), there is no explanation as to why ‘Man/Genderqueer’ was chosen as the index variable, or what this category means. Something has to be the index, of course, but what was the reasoning behind this choice? Are male and genderqueer somehow synonymous? Since the analysis defines men as the standard against which other categories are measured, it requires some explanation.
Finally in terms of presentation issues, I found the tables quite overwhelming. Is it necessary, for instance, to report each year of age? Could these not be grouped? We know that the analysis will be affected by the grouping, but still, this is a lot of raw data to assimilate, and not very meaningful. Present categories, or mean-median-mode with SD and that should be enough for a table. Transparency is good, but raw data is the domain of the researchers, not readers. I strongly encourage the authors to consider reducing the number and size of the tables, breaking them up so that these are separate tables, limiting tables to four or five necessary ones, and only presenting what data support the thesis or analysis, rather than overwhelming the reader with raw data. As above, raw data is the domain of the researchers, not the reader. Or put the data in a data repository.
Now to a couple of more substantive issues. The paper reports that 59.9% of participants had received a prior mental health diagnosis and that 69.2% reported that their mental health was only poor or fair. This is remarkable. What do these data mean? Are they suggestive of overall Canadian young people? Much of this (43.4%) was anxiety: is there a trend in Canada to over-diagnose anxiety, or is there an epidemic of anxiety in young people? This is unlikely to be an artifact of Covid, since the question was about pre-existing diagnoses—surely a finding this large merits at least a brief explanation even before the discriminate analysis. Then the reader is prompted to wonder, if these were pre-existing diagnoses (presumably not self-diagnoses), what was the follow-up plan by the mental health provider? That seems like a significant indictment of Canadian mental health services. The paper calls for greater access and reduced barriers, but there is no clear statement about what this improved access might look like. Participants were mostly recruited from social media, so they are clearly connected, and nearly a third of participants reported illicit substance use in the past year, so they clearly get out and about. What are the barriers, and how can they be overcome? The paper only suggests that barriers were associated with educational attainment. We know that gender and sexually diverse young people can exclude themselves from formal education early if that education is not specifically inclusive: are there implications in this study for education as well as mental health providers?
All of that said, the Discussion section is thorough and wide-ranging. It would be helpful to work across disciplines in a future paper so that more specific recommendations could be made to providers of mental health services.
The references are relevant and recent.
Minor issues
- In lines 179 and 187 the authors are commended for including the threshold of significance, but (a small point) is this not usually expressed as α[alpha]=0.05?
- The authors will also want to go through and ensure that the numerical data are consistently aligned throughout (flush right is preferred, I believe, rather than centred.) In Table 4 consistency in the decimal expressions (to both the left and right of the decimal) would make the table more readable (PTSD percentage is expressed as 12, for instance, rather than 12.0), as would ensuring that numerical expressions in the text are also consistent (see, for instance ‘.1%’ in line 237—it would be more readable as ‘0.1%’). It is also usual to avoid beginning a sentence with a number (see line 231).
- Some of the table rows are mis-aligned, and the authors will wish to ensure this is tidied up. The variable column is also very difficult to read as aligned (What is ‘Heterose’? Oh, wait there’s another line, ‘xual/’). Minor issues to correct but it will improve readability considerably. One way to manage this might be to reduce the category labels: if the larger category is ‘gender’ then the next level of label only needs to say ‘fluid’ ‘queer’, ‘non-binary’ etc., rather than genderfluid, genderqueer, etc.
- There is no apparent order to the way the rows are presented in the tables: It is neither greatest to smallest (which would be best), nor alphabetical. Please be friendly to your readers and present the data categories consistently.
- It would be good to ensure that tables do not break across pages by reducing the number and size of tables. Make some of these separate tables rather than one (very) long table. Where tables breaking across pages is inevitable, it would be good to repeat the heading column on any new page. I found myself jumping back and forth to understand what I was looking at.
- In line 329 we read that 59.9% of respondents expressed suicidal ideation, but previously (Table 2, third category and line 224) that number is 62.3%.
- In line 330, consider ‘A study conducted in 2019, prior to the COVID-19 pandemic, reported…’—perhaps a little less awkward.
